# Metal-Based Scaffolds of Schiff Bases Derived from Naproxen: Synthesis, Antibacterial Activities, and Molecular Docking Studies

**DOI:** 10.3390/molecules24071237

**Published:** 2019-03-29

**Authors:** Muhammad Ashraf Shaheen, Shanshan Feng, Mehwish Anthony, Muhammad Nawaz Tahir, Mubashir Hassan, Sung-Yum Seo, Saeed Ahmad, Mudassir Iqbal, Muhammad Saleem, Changrui Lu

**Affiliations:** 1Key Laboratory of Science and Technology of Eco-Textiles, Ministry of Education, College of Chemistry, Chemical Engineering and Biotechnology, Donghua University, Shanghai 201620, China; mashaheen@uos.edu.pk (M.A.S.); sandyfss@163.com (S.F.); 2Department of Chemistry, University of Sargodha, Sargodha 40100, Pakistan; mehwish.anthony.chemist@gmail.com; 3Department of Physics, University of Sargodha, Sargodha 40100, Pakistan; dmntahir_uos@yahoo.com; 4Collage of Natural science, Department of Biology, Kongju National University, Gongju, Chungcheongnam 32588, Korea; 18817326450@163.com (M.H.); 18226212391@163.com (S.-Y.S.); 5Department of Chemistry, College of Sciences and Humanities Prince Sattam bin Abdul Aziz University, Al-Kharj 11942, Saudi Arabia; saeed_a786@hotmail.com; 6Department of Chemistry, School of Natural Sciences (SNS), National University of Sciences & Technology (NUST), H-12, Islamabad 44000, Pakistan; mudassir.iqbal@sns.nust.edu.pk; 7Department of Chemistry, University of Sargodha, Sub-campus, Bhakkar 30000, Pakistan

**Keywords:** naproxen, Schiff base, hydrazide, complexation, bioactivity

## Abstract

We report here the synthesis, characterization, and antibacterial evaluation of transition metal complexes of Ni, Cu, Co, Mn, Zn, and Cd (**6a**–**f**), using a Schiff base ligand (**5**) derived from naproxen (an anti-inflammatory drug) and 5-bromosalicylaldehyde by a series of reactions. The ligand and the synthesized complexes were characterized by elemental analysis, UV-Visible, FTIR, and XRD techniques. The ligand **5** behaves as a bidentate donor and coordinates with metals in square planar or tetrahedral fashion. In order to evaluate its bioactivity profile, we screened the Schiff base ligand and its metal complexes (**6a**–**f**) against different species of bacteria and the complexes were found to exhibit significant antibacterial activity. The complexes showed more potency against *Bacillus subtilis* as compared to the other species. Moreover, we modeled these complexes’ binding affinity against COX1 protein using computational docking.

## 1. Introduction

The therapeutic usage of metal-based substances dates back to the ancient Greeks, Assyrians, Egyptians, and Chinese, who have been known to use cinnabar and various compounds of gold, silver, and bismuth to cure different diseases. In traditional Chinese medicine, arsenic drugs, such as arsenic trioxide (ATO), were used as antiseptic and anticancer agents or in the treatment of rheumatoid diseases, syphilis, and psoriasis. Past research has identified a large number of metal complexes for their potential use as therapeutics and catalysts. Since the 16th century, transitions metal complexes have shown effective therapeutic powers in treating cancer, diabetes, and arthritis. More recently, studies have shown that metal complexes, such as cisplatin and its analogues, threaten various types of cancer [1,2].

The ligand plays a vital role in determining the properties of the metal complexes for a specific purpose. Therefore, a variety of ligands with different donor sites have formed metal-based scaffolds for therapeutic and diagnostic agents [3]. Among them, Schiff bases, with different donor sites with chelation tendency toward several metal ions [4,5], proved to be versatile ligands used in the medical field, including diagnostics [6]. For example, nonsteroidal anti-inflammatory drugs (NSAIDs) and their derivatives have shown structural and physicochemical diversity and biocompatibility as ligands, forming transition metal complexes endowed with therapeutic effects [7,8,9,10]. Naproxen, ibuprofen, and aspirin are the most prominent NSAIDs that can form metal complexes as ligands due to their medicinal effects and biocompatibility [8,9,10,11].

To obtain good drug candidates, the investigation of bioactivities of the well-known NSAIDs and their hybridized compounds via Schiff base formation takes priority. Studies have shown that the NSAID complexes with transition metals can reduce gastrointestinal mucosal disorders effectively [12]. Several NSAIDs metal complexes possess anti-oxidative features and anti-inflammatory activities, such as organo-tin compounds [13,14,15,16,17]. Reports show that tolfenamic acid–Co (II) complex interacts with albumin serum and DNA [16,18,19]. Meanwhile, the naproxen and diclofenac acid form complexes with Cu (II) ions that exhibit antimicrobial and anti-tumor activities [20]. Ibuprofen forms complexes with Cu (II) and Zn (II) ions and shows inhibitory growth and antibacterial effects [21,22,23,24].

Another class of organic molecules, the hydrazides, show diverse properties, such as anticonvulsant, anti-inflammatory, antibacterial, antimalarial, antifungal, and anti-tuberculosis activities [25,26,27]. The hydrazides and acyl-hydrazide complexes of naproxen have shown antimalarial [28,29], anti-tumor, anti-inflammatory (Zn, Co, and Ca) [30,31], and anti-oxidative effects (Co complex) [32]. Consequently, we synthesized Naproxen hydrazides and reacted them with salicylic aldehydes to obtain the Schiff bases, which were then utilized to obtain the metal complexes. The resulting metal complexes were characterized by different spectroscopic techniques and screened for antibacterial activity, yielding several potential candidate compounds with antibacterial activity.

## 2. Results and Discussion

### 2.1. Synthesis and Structure of the Schiff Base (**5**)

Compound **5**, the Schiff base of hydrazide, was prepared as illustrated in Scheme 1. Briefly, sodium salt of naproxen was treated with HCl to form 2-(6-methoxynaphthalen-2-yl) propanoic acid and was subsequently converted to methyl 2-(6-methoxynaphthalen-2-yl) propanoate. The formation of ester was confirmed by the appearance of an IR absorbance peak at 1083.99 cm^−1^ for (OC−OCH_3_). Methyl ester, **3**, was reacted with hydrazine to prepare the hydrazide, and then treated further with 5-Bromosalicylaldehyde to prepare the Schiff base ligand **5**. The FTIR spectrum of the Schiff base shows peaks for (N−H) at 3226.91 cm^−1^ for (C=N) at 1658.78 cm^−1^, for (NC=O) at 1614.42 cm^−1^, for (Ar−OH) at 3541.31cm^−1^, for (C−O) at 1203.58 cm^−1^, for (Ar−OCH_3_) at 1271.09 cm^−1^, for (OH) at 3049.46 cm−1, and for (C−Br) at 623.01 cm^−1^. The compound **5** exhibits λmax at (C=N) 282.0 nm. The FTIR and UV data agree with the proposed structure of **5** [33,34,35,36].

### 2.2. Single Crystal XRD Analysis for the Schiff Base (**5**)

In compound **5**, (N’-[(*E*)-(5-bromo-2-hydroxyphenyl)methylidene]-2-(6-methoxynaphthalen-2-yl) propanehydrazide), the aldehyde moiety (C1-C7/O1/Br1), A and the naphthalene moiety (C11–C20), B share the same plane with r.m.s. deviations of 0.0161 and 0.0131 Å, respectively. The hydrazidal (C8/C9/N1/N2/O2), part C is roughly planar with a root-mean-square (r.m.s.) deviation of 0.2889 Å. The dihedral angle between A/B, A/C, and B/C is 60.52 (11)°, 49.88 (15)°, and 45.42 (17)°, respectively. Figure 1 shows intramolecular H-bonding of O-H…N type with an S (6) loop. Intramolecular interactions link the molecules due to N-H…O interactions between the imine and carbonyl groups in the form of C (4) chains along the crystallographic a-axis. The molecules have pi-pi stacking between the similar benzene rings in the range [3.131(2)–3.131(2) Å] and have the slippage range [3.572–3.786 Å] (Figure 2). The entire crystal data have been tabulated in Table 1.

The crystal structure data have been deposited at the Cambridge Crystallographic Data Centre under CCDC No. 1556675. These data can be obtained free of charge via http://www.ccdc.cam.ac.uk/conts/retrieving.html (or from the CCDC, 12 Union Road, Cambridge CB2 1EZ, UK; Fax: +44 1223 336033; E-mail: deposit@ccdc.cam.ac.uk). 

### 2.3. Synthesis of Target Molecules **6a**–**f**

The Schiff base ligand **5** solution was treated with a corresponding metal salt solution and colored complexes were obtained as precipitates. The color change in the reaction mixture before and after induction of metal ions indicates complex formation. Thin layer chromatography was used to verify the formation of target molecules **6a**–**f**. The FTIR spectral analysis registered the disappearance of the signal corresponding to the phenolic hydroxyl stretching vibration in ligand **5**, caused by the chelation with corresponding metal ions. Moreover, the signal due to C=N stretching vibration shifted. Scheme 1 summarizes the synthetic protocol adopted to synthesize the target molecules **6a**–**f**.

### 2.4. Anti-Bacterial Activity

We then screened all of the abovementioned compounds, i.e., Schiff bases and their metal complexes, for their antibacterial activities [37], showing that the Schiff base ligand and its metal complexes exhibited significant antibacterial activities. We selected four stains of different types of bacterial species, *Escherichia coli, Streptococcus aureous, Salmonella typhae,* and *Bacillus subtilis*, for use in evaluating antimicrobial potential by the diffusion method (Table 2). Briefly, autoclaved glass plates were used for conducting bacterial activity. Distilled water was used for preparing a solution of agar gel and Ciprofloxacin was used as a reference. Samples of all the derivatives of naproxen were placed in bacterial medium for 24 h. The activity was measured in terms of inhibition zones in mm. All compounds tested showed inhibition to bacteria growth, comparable with that of ciprofloxacin. The complexes with Co, Cu, and Co in general showed improvements over the free ligand across all four bacterial species.

### 2.5. Computational Analysis of Synthesized Compounds Against COX1

#### 2.5.1. Molecular Docking Analyses

Prostaglandin endoperoxide H synthases (PGHSs)-1 and -2 (also called cyclooxygenases (COXs)-1 and -2) catalyze the committed step in prostaglandin biosynthesis. In the pathway, NSAIDs target COX1 and COX2 [38]. Human COX1 enzyme comprises two chains with 553 residues. The docked complexes of synthesized compounds **6a**–**f** bound to COX1 were analyzed on the basis of the lowest binding energy values (kcal/mol) and hydrogen/hydrophobic bonding analyses. The analysis showed that all compounds bind to the enzyme with binding energy values around −11.70 kcal/mol, compared to the reference structure energy value (−8.20 kcal/mol).

#### 2.5.2. Structure Activity Relationship (SAR) Analyses of Synthesized Compounds and Target Protein

The SAR analyses showed that compounds (**6a**–**f**) directly interact within the active region of COX1. Results showed that compounds build hydrogen and hydrophobic interactions with active site target residues with appropriate binding distances. In the 6a-docking results, we observed two interactions at different binding residues. The carbonyl oxygen atom of **6a** interacts with Asn122 and forms a hydrogen bond having a length of 3.10 Å. Another hydrophobic interaction was observed against Leu112 with bond length 4.26 Å. In **6b**-docking complex, we observed two hydrophobic interactions against target protein. In **6b**-docking complex, we observed a weak binding interaction at Arg79 and Tyr64 residues having the bond lengths 5.07 and 4.69 Å, respectively. In both docking complexes, the metals containing ligands (**6a** and **6b**) were different; therefore, our model shows little variant binding interaction pattern.

As in **6a**-docking, in **6c**-docking complex two hydrophobic interactions and one hydrogen bond were observed at different residues (Pro84, Arg79, and Lys532) of target protein. The ligand **6c** forms hydrophobic interactions having a bond length of 4.95 and 3.13 Å. Similarly, we show a weak hydrogen bond with bond length 5.07 Å. In **6d**-COX1 docking, a couple of hydrogen bonds were seen at Asn122 and Ser126 at a bond length of 2.91 and 2.82 Å, respectively. In both **6c** and **6d** docking complexes, the hydrogen bonds strengthen the docking complexes. In **6e** and **6f** complexes, we observed hydrophobic interactions against the target protein residues Thr80, Arg120, Ile89, Tyr64, and Ile46, with distances of 2.96, 3.91, 5.22, 5.06, and 4.37 Å, respectively (Figure 3). The comparative results showed that the binding residues among standard compounds (**5**) and other derivatives were similar in **6a** and **6d**, whereas little fluctuations in residues exist in other docking complexes. A literature study also justified that these binding pocket residues are significant in downstream signaling pathways [39,40,41].

## 3. Materials and Methods

The chemicals used were of analytic grades, purchased from Sigma Aldrich, Pakistan, and used without purification except where mentioned. Sodium salt of naproxen was received from Moringa Pharma, Pakistan, as a gift. FTIR Prestige−21 (Shimadzu, Kyoto, Japan) was used for FTIR spectra of the synthesized products. Elemental analyses were carried out with an Exeter Analytical Inc-CE-440 Elemental Analyzer. Melting points were determined by a Gallenkamp digital melting point apparatus (MFB−595−010M). Bruker (KAPPA Apex II) XRD was used for determination of crystalline structure. ^1^H-NMR and ^13^C-NMR spectra were recorded on a Varian Unity INOVA (300 MHz) spectrometer. ^1^H-NMR (300 MHz) and ^13^C-NMR (75 MHz) chemical shift values are reported as δ using the residual solvent signal as an internal standard. All NMR measurements were recorded in CDCl_3_ as a solvent unless mentioned otherwise. Antibacterial activity was screened by using fresh cultures of different species of bacteria by following the disc diffusion method [42].

### 3.1. Synthesis of 2-(6-Methoxynaphthalen-2-yl)propanoicacid (**2**)

Naproxen, (**2**) was prepared by treating equimolar amount of Naproxen-Na, (**1**) and HCl.

Yield: 98%, mp. 152 °C. ῡ cm^−1^: (C=O) 1718 cm^−1^, (C−O) 1174 cm^−1^; (O−H) 3186 cm^−1^, (Ar−OCH3) 1265 cm^−1^. C_14_H_14_O_3_ (230.3): calcd. C, 73.03; H, 6.13; O, 20.85; found C, 72.93; H, 6.03.

### 3.2. Synthesis of 2−(6−Methoxynaphthalen−2−yl)propanehydrazide (**4**)

The hydrazide of the methyl ester (**4**), was prepared from naproxen (**2**), through a series of reactions depicted in Scheme 1 according to reported [43] procedure with some modifications.

### 3.3. Synthesis of (N’-[(E)-(5-bromo-2-hydroxyphenyl) methylidene]-2-(6-methoxynaphthalen-2-yl)propanehydrazide) (**5**)

A methanolic solution of naproxen hydrazide, (**4**) (2.00g, 8.2 mmol) was added to the acidified methanolic solution of 5-bromosalicyladehyde (1.65g, 8.2 mmol) and the mixture was refluxed for 4−5 h. The reaction mixture was concentrated and filtered. The product was washed with pet. ether repeatedly and recrystallized from methanol and *n*-hexane. Yield: 3.82g: 76%, mp. 138 °C. ῡ cm^−1^: (N−H) 3226.91 cm^−1^, (C=N) 1658.78 cm^−1^, (NC=O) 1614.42 cm^−1^, (Ar−OH) 3541.31 cm^−1^, (C−O) 1203.58 cm^−1^, (Ar−OCH_3_) 1271.09 cm^−1^, (OH) 3049.46 cm^−1^, (C−Br) 623.01 cm^−1^. λ max: (C=N) 282.0 nm. C_21_H_19_BrN_2_O_3_ (427.3): calcd. C, 59.03; H, 4.48; Br, 18.70; N, 6.56; O, 11.23; found C, 58.73; H, 4.18; Br, 18.30; N, 6.16. ^1^H-NMR: δ 1.41 (d, 3H, Me), 3.83 (s, 3H, -OMe), 3.53 (q, 1H, -CH), 3.76 (m, 2H, -NH_2_), 6.41 (d, 1H, -ArH), 7.16–7.20 (m, 2H, -ArH), 7.37 (d, 1H, -ArH), 7.41–7.44 (m, 2H, -ArH), 7.81 (s, 1H, -ArH), 7.86–7.88 (m, 2H, -ArH), 8.68 (s, 1H, =CH). ^13^C-NMR: δ 15.2, 45.4, 55.8, 105.4, 110.8, 118.5, 119.6, 120.8, 126.0, 126.7, 128.3, 128.9, 129.4, 132.3, 132.9, 133.4, 135.3, 146. 3, 169.7, 156.1, 177.7.

### 3.4. Synthesis of Metal Complexes **6a**-**f** from Schiff Base Ligand (**5**)

Complexes **6a**–**f** were prepared by the same general procedure. The methanolic solution of the ligand (**5**) and the metal salts were mixed in a 2:1 ratio in the presence of 10% Na_2_CO_3_ and refluxed for 1–2 h. The reaction mixture was concentrated and filtered. The solid products were washed and recrystallized with appropriate solvents.

#### 3.4.1. Synthesis of Ni Complex (**6a**)

Schiff base ligand (**5**) (0.785 g, 1.8 mmol) was added to methanol (25 mL) in a round bottom flask (100 mL) and heated for 30 min with continuous stirring. A 10% NaHCO_3_ (2 mL) solution was added to the reaction mixture. The methanolic solution of Ni(CH_3_COO)_2_·4H_2_O (0.228 g, 0.91 mmol) was added to the Schiff base mixture. The reaction mixture was refluxed for 2 h with continuous stirring. The light green color ppt. was washed with distilled water repeatedly and dried. **Yield**: 0.801g; 97%, mp. 298 °C FTIR Peaks: (N−H) 3184 cm**^−^**^1^, (Ni−N) 449 cm^−1^, (Ni−O) 480 cm^−1^, (C=N) 1604 cm^−1^, (Ar−OCH_3_) 1230 cm^−1^, (C−Br) 565 cm^−1^, (CO) 1170 cm^−1^, (NC=O) 1749 cm^−1^. λ max: (C=N) 394.0 nm. C_42_H_36_N_4_NiO_6_ (911.3): calc. C, 55.36; H, 3.98; N, 6.15; found C, 55.06; H, 3.68; N, 5.9.

#### 3.4.2. Synthesis of Mn (II) Complex (**6b**)

The maroon color ppt. was rinsed with distilled water repeatedly: Yield: 0.275 g; 84%, mp. 270 °C FTIR Peaks: (N−H) 3165 cm^−1^, (Mn−N) 428 cm^−1^, (Mn−O) 480 cm^−1^, (C=N) 1604 cm^−1^, (Ar−OCH_3_) 1219 cm^−1^, (C−Br) 536 cm^−1^, (CO) 1174 cm^−1^, (NC=O) 1724 cm^−1^. λ max: (C=N) 404.0 nm. C_42_H_36_Br_2_MnN_4_O_6_ (907.5): calcd. C, 55.59; H, 4.00; N, 6.17; found C, 55.39; H, 3.90; N, 6.07.

#### 3.4.3. Synthesis of Cu Complex (**6c**)

The green ppt. was washed with distilled water several times: Yield: 0.277 g; 84%, mp. 265 °C FTIR Peaks: (N−H) 3224 cm^−1^, (Cu−N) 470 cm^−1^, (Cu−O) 542 cm^−1^, (C=N) 1612 cm^−1^, (Ar−OCH_3_) 1276 cm^−1^, (C−Br) 648 cm^−1^, (CO) 1190 cm^−1^, (NC=O) 1879 cm^−1^; λ max: (C=N) 316 nm. C_42_H_36_Br_2_CuN_4_O_6_: calcd. C, 55.06; H, 3.96; N, 6.12; found C, 54.96; H, 3.76; N, 5.94.

#### 3.4.4. Synthesis of Zn Complex (**6d**)

Pale yellow color ppt. was obtained after filtration and washing with water: Yield: 0.955 g: 89%, mp. 300+ °C.FTIR Peaks: (N−H) 3116 cm^−1^, (Zn−N) 457 cm^−1^, (Zn−O) 476 cm^−1^, (C=N) 1616 cm^−1^ (Ar−OCH_3_) 1244 cm^−1^, (C−Br) 551 cm^−1^, (CO) 1165cm^−1^, (NC=O) 1739 cm^−1^; λ max: (C=N) 301.0 nm. C_42_H_36_Br_2_N_4_O_6_Zn (917.9): calcd. C, 54.95; H, 3.95; N, 6.10; found C, 54.75; H, 3.65; N, 5.90.

#### 3.4.5. Synthesis of Co Complex (**6e**)

After performing filtration, orange color ppt. was washed with distilled water thoroughly: Yield: 0.733 g; 69%, mp. 279 °C; FTIR Peaks: (N−H) 3138 cm^−1^, (Co−N) 460 cm^−1^, (Co−O) 470 cm^−1^, (C=N) 1608 cm^−1^, (Ar−OCH_3_) 1286 cm^−1^, (C−Br) 551 cm^−1^, (CO) 1168 cm^−1^, (NC=O) 1816 cm^−1^; λ max: (C=N) 300.0 nm. C_42_H_36_Br_2_CoN_4_O_6_ (911.5): calcd. C, 55.34; H, 3.98; N, 6.15; found C, 55.04; H, 3.78; N, 6.01.

#### 3.4.6. Synthesis of Cd Complex (**6f**)

After filtration, creamy white color ppt. was rinsed repeatedly with Pet. Ether: Yield: 0.130 g; 92%, mp. 293 °C FTIR Peaks: (N−H) 3132 cm^−1^, (Cd−N) 555 cm^−1^, (Cd−O) 603 cm^−1^, (C=N) 1612 cm^−1^, (Ar−OCH_3_) 1244 cm^−1^, (C−Br) 636 cm^−1^, (CO) 1170 cm^−1^, (NC=O) 1776 cm^−1^; λ max: (C=N) 299 nm. C_42_H_36_Br_2_CdN_4_O_6_ (965.0): calcd. C, 52.28; H, 3.76; N, 5.81; found C, 52.01; H, 3.46; N, 5.41.

FTIR spectra of the compounds can be obtained from the corresponding author free of cost on request.

### 3.5. Anti-Bacterial Testing

Autoclaved glass plates were used for conducting bacterial activity. Distilled water was used for preparing a solution of agar gel and Ciprofloxacin was used as a reference. Samples of all the derivatives of naproxen were placed in bacterial medium for 24 h. The activity was measured in terms of inhibition zones in mm.

### 3.6. Molecular Docking

#### 3.6.1. Retrieval of COX1 Structure from the Protein Data Bank (PDB)

The three-dimensional (3D) crystal structure of COX1 (PDBID: 3N8Z) was accessed from the PDB (http://www.rcsb.org). The retrieved protein structure was further minimized by using a conjugate gradient algorithm and amber force field in UCSF Chimera 1.10.1 [44].

#### 3.6.2. Designing of Ligands and Molecular Docking

The synthesized ligands (**6a**–**f**) were sketched in the ACD/ChemSketch tool and minimized by UCSF Chimera 1.10.1. A docking experiment was used on all synthesized compounds (**6a**–**f**) COX1 using a PyRx docking tool [45]. In the docking experiment, grid box parametric dimension values were adjusted as X = −25.6989, Y = 57.7957, and Z = 8.339, respectively. The default exhaustiveness = 8 value was used to obtain the finest binding conformational pose of protein–ligand docked complexes. The docked complexes were evaluated on lowest binding energy (Kcal/mol) values, and the hydrogen/hydrophobic interactions pattern using Discovery Studio (4.1) and UCSF Chimera 1.10.1. To check the validity of our docked complexes, a reference structure (Comp. 5) was also docked to check their binding pattern against COX1.

## 4. Conclusions

In this study, we synthesized metal derivatives of Naproxen-based Schiff bases in acidic or basic medium as required. We characterized the prepared compounds, namely naproxen acid from naproxen salt, methyl ester, hydrazide, Schiff base, and metal complexes, by different spectroscopic techniques. Subsequently, we screened these compounds for in vitro antibacterial activity, revealing that all complexes possess such activity. Lastly, we proved antibacterial activity through a computational approach. Our results suggest that these compounds inhibit Cyclooxygenase-1 through specific binding to its active sites. We calculated their individual binding energy against the target protein. Metal complexes with Co, Cu, and Co showed higher antibacterial activity than ciprofloxacin across all four bacterial species.

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
