# Peer review of "Metal-Based Scaffolds of Schiff Bases Derived from Naproxen: Synthesis, Antibacterial Activities, and Molecular Docking Studies"

_molecules, 2019, doi:10.3390/molecules24071237_

Round 1

Reviewer 1 Report

I have no particular suggestions to make, except for these amendments:

page 2, line 46

"...Schiff bases having different..." instead of "...Schiff bases have different ..."

page 2, line 56

"...transition metals can effectively ..." instead of "...transition metals and can effectively ..."

page 2, line 59

"... the naproxen and diclofenac acid form complexes ..." instead of " ... the naproxen and diclofenac acid forms complexes ..."

page 2, line 60

"... with Cu (II) ions that exhibit ..." instead of "... with Cu (II) ions exhibit ..."

page 2, line 67

" ... Naproxen hydrazides and submitted them to reaction with ..." instead of " ... Naproxen hydrazides and reacted with ..."

page 2, line 73

"...was prepared as illustrated in figure 3." instead of "... was prepared as illustrated in figure 1."

page 8, line 179

"... was prepared from naproxen (2), through a series of ..." instead of "... was prepared from naproxen, (2) through a series of ..."

page 9, line 241

"... was accessed from the protein data bank ..." instead of "... was accessed form the protein data bank ..."

page 9, line 249

"... obtain the finest binding ..." instead of "... obtained the finest binding ..."

Author Response

we have done the changes mentioned by reviewer 1

Reviewer 2 Report

- The entitled manuscript is a good starting point for future studies. In this context, I believe that a more comprehensive study addressing the single crystal X-ray diffraction of the studied complexes would have helped to achieve better conclusions regarding the intermolecular interactions between complexes and COX1. However my mayor concern is about the tetrahedral geometry that authors attribute to the metal complexes. It is not clear for me how the authors conclude that statement reading the manuscript. The authors should clarify this point. In addition, do the models conclude a square planar geometry to the complexes?? This fact needs to be clarified by the authors too. 

- pi,pi stacking interactions should be considered tacking into account the chemical structure of the ligand.

- The analysis of molecular recognition in modelling could be broader.

-Please check references  3, 22, 31, 37 and 42. Information is missing.

Author Response

- The entitled manuscript is a good starting point for future studies. In this context, I believe that a more comprehensive study addressing the single crystal X-ray diffraction of the studied complexes would have helped to achieve better conclusions regarding the intermolecular interactions between complexes and COX1.

We thank the reviewers for the support. We also agree that XRD would be much more informative. We would proceed with this in our future study for sure.

However my mayor concern is about the tetrahedral geometry that authors attribute to the metal complexes. It is not clear for me how the authors conclude that statement reading the manuscript. The authors should clarify this point. In addition, do the models conclude a square planar geometry to the complexes?? This fact needs to be clarified by the authors too.

We thank the reviewers for the constructive comments. Since Ni, Cu, Co, Mn, Zn, and Cd commonly adopt either square planar or tetrahedral coordination geometry, our Schiff base complex adopt one of the two forms. However, due to steric hindrance we speculate that tetrahedral geometry is more likely. We have modified our text to reflect this conclusion.

- pi,pi stacking interactions should be considered tacking into account the chemical structure of the ligand.

We have modified our text according to recommended modifications.

- The analysis of molecular recognition in modelling could be broader.

We have extended our analysis on modelling according to recommended modifications.

-Please check references  3, 22, 31, 37 and 42. Information is missing.

We have fixed the abovementioned references according to recommended modifications.